# FasterRisk: Fast and Accurate Interpretable Risk Scores

**Jiachang Liu**[1]*   **Chudi Zhong**[1]*   **Boxuan Li**[1]   **Margo Seltzer**[2]   **Cynthia Rudin**[1]

[1] Duke Univeristy [2] University of British Columbia

{jiachang.liu, chudi.zhong, boxuan.li}@duke.edu
mseltzer@cs.ubc.ca, cynthia@cs.duke.edu

## Abstract

Over the last century, *risk scores* have been the most popular form of predictive model used in healthcare and criminal justice. Risk scores are sparse linear models with integer coefficients; often these models can be memorized or placed on an index card. Typically, risk scores have been created either without data or by rounding logistic regression coefficients, but these methods do not reliably produce high-quality risk scores. Recent work used mathematical programming, which is computationally slow. We introduce an approach for efficiently producing a collection of high-quality risk scores learned from data. Specifically, our approach produces a pool of almost-optimal sparse continuous solutions, each with a different support set, using a beam-search algorithm. Each of these continuous solutions is transformed into a separate risk score through a "star ray" search, where a range of multipliers are considered before rounding the coefficients sequentially to maintain low logistic loss. Our algorithm returns all of these high-quality risk scores for the user to consider. This method completes within minutes and can be valuable in a broad variety of applications.

## 1   Introduction

*Risk scores* are sparse linear models with integer coefficients that predict risks. They are arguably the most popular form of predictive model for high stakes decisions through the last century and are the standard form of model used in criminal justice [4, 22] and medicine [19, 27, 34, 31, 41].

Their history dates back to at least the criminal justice work of Burgess [8], where, based on their criminal history and demographics, individuals were assigned integer point scores between 0 and 21 that determined the probability of their "making good or of failing upon parole." Other famous risk scores are arguably the most widely-used predictive models in healthcare. These include the APGAR score [3], developed in 1952 and given to newborns, and the $CHADS_2$ score [18], which estimates stroke risk for atrial fibrillation patients. Figures 1 and 2 show example risk scores, which es-

| 1. | Oval Shape | -2 points | | ... |
| 2. | Irregular Shape | 4 points | + | ... |
| 3. | Circumscribed Margin | -5 points | + | ... |
| 4. | Spiculated Margin | 2 points | + | ... |
| 5. | Age $\geq$ 60 | 3 points | + | ... |
| | | **SCORE** | = | |

| SCORE | -7 | -5 | -4 | -3 | -2 | -1 |
|---|---|---|---|---|---|---|
| **RISK** | 6.0% | 10.6% | 13.8% | 17.9% | 22.8% | 28.6% |
| **SCORE** | 0 | 1 | 2 | 3 | 4 | $\geq$ 5 |
| **RISK** | 35.2% | 42.4% | 50.0% | 57.6% | 64.8% | 71.4% |

Figure 1: Risk score on the mammo dataset [15], whose population is biopsy patients. It predicts the risk of malignancy of a breast lesion. Risk score is from FasterRisk on a fold of a 5-CV split. The AUCs on the training and test sets are 0.854 and 0.853, respectively.

---

*These authors contributed equally.

timate risk of a breast lesion being malignant.

Risk scores have the benefit of being easily memorized; usually their names reveal the full model – for instance, the factors in CHADS$_2$ are past **C**hronic heart failure, **H**ypertension, **A**ge$\geq$75 years, **D**iabetes, and past **S**troke (where past stroke receives **2** points and the others each receive 1 point). For risk scores, counterfactuals are often trivial to compute, even without a calculator. Also, checking that the data and calculations are correct is easier with risk scores than with other approaches. In short, risk scores have been created by humans for a century to support a huge spectrum of applications [2, 23, 30, 43, 44, 47], because humans find them easy to understand.

| 1. | Irregular Shape | 4 points | | ... |
|----|-----------------|----------|-----|-----|
| 2. | Circumscribed Margin | -5 points | + | ... |
| 3. | SpiculatedMargin | 2 points | + | ... |
| 4. | Age $\geq$ 45 | 1 point | + | ... |
| 5. | Age $\geq$ 60 | 3 points | + | ... |
| | | **SCORE** | = | |

| SCORE | -5 | -4 | -3 | -2 | -1 | 0 |
|-------|------|------|-------|-------|-------|-------|
| RISK | 7.3% | 9.7% | 12.9% | 16.9% | 21.9% | 27.8% |
| SCORE | 1 | 2 | 3 | 4 | 5 | 6 |
| RISK | 34.6% | 42.1% | 50.0% | 57.9% | 65.4% | 72.2% |

Figure 2: A second risk score on the mammo dataset on the same fold as in Figure 1. The AUCs on the training and test sets are 0.855 and 0.859, respectively. FasterRisk can produce a diverse pool of high-quality models. Users can choose a model that best fits with their domain knowledge.

Traditionally, risk scores have been created in two main ways: (1) without data, with expert knowledge only (and validated only afterwards on data), and (2) using a semi-manual process involving manual feature selection and rounding of logistic regression coefficients. That is, these approaches rely heavily on domain expertise and rely little on data. Unfortunately, the alternative of building a model *directly* from data leads to computationally hard problems: optimizing risk scores over a global objective on data is NP-hard, because in order to produce integer-valued scores, the feasible region must be the integer lattice. There have been only a few approaches to design risk scores automatically [5, 6, 9, 10, 16, 32, 33, 38, 39, 40], but each of these has a flaw that limits its use in practice: the optimization-based approaches use mathematical programming solvers (which require a license) that are slow and scale poorly, and the other methods are randomized greedy algorithms, producing fast but much lower-quality solutions. We need an approach that exhibits the best of both worlds: speed fast enough to operate in a few minutes on a laptop and optimization/search capability as powerful as that of the mathematical programming tools. Our method, FasterRisk, lies at this intersection. It is fast enough to enable interactive model design and can rapidly produce a large pool of models from which users can choose rather than producing only a single model.

One may wonder why simple rounding of $\ell_1$-regularized logistic regression coefficients does not yield sufficiently good risk scores. Past works [37, 39] explain this as follows: the sheer amount of $\ell_1$ regularization needed to get a very sparse solution leads to large biases and worse loss values, and rounding goes against the performance gradient. For example, consider the following coefficients from $\ell_1$ regularization: [1.45, .87, .83, .47, .23, .15, ... ]. This model is worse than its unregularized counterpart due to the bias induced by the large $\ell_1$ term. Its rounded solution is [1,1,1,0,0,0,..], which leads to even worse loss. Instead, one could multiply all the coefficients by a constant and then round, but which constant is best? There are an infinite number of choices. Even if some value of the multiplier leads to minimal loss due to rounding, the bias from the $\ell_1$ term still limits the quality of the solution.

The algorithm presented here does not have these disadvantages. The steps are: (1) Fast subset search with $\ell_0$ optimization (avoiding the bias from $\ell_1$). This requires the solution of an NP-hard problem, but our fast subset selection algorithm is able to solve this quickly. We proceed from this accurate sparse continuous solution, preserving both sparseness and accuracy in the next steps. (2) Find a pool of diverse continuous sparse solutions that are almost as good as the solution found in (1) but with different support sets. (3) A "star ray" search, where we search for feasible integer-valued solutions along multipliers of each item in the pool from (2). By using multipliers, the search space resembles the rays of a star, because it extends each coefficient in the pool outward from the origin to search for solutions. To find integer solutions, we perform a local search (a form of sequential rounding). This method yields high performance solutions: we provide a theoretical upper bound on the loss difference between the continuous sparse solution and the rounded integer sparse solution.

Through extensive experiments, we show that our proposed method is computationally fast and produces high-quality integer solutions. This work thus provides valuable and novel tools to create risk scores for professionals in many different fields, such as healthcare, finance, and criminal justice.

**Contributions**: Our contributions include the three-step framework for producing risk scores, a beam-search-based algorithm for logistic regression with bounded coefficients (for Step 1), the search algorithm to find pools of diverse high-quality continuous solutions (for Step 2), the star ray search technique using multipliers (Step 3), and a theorem guaranteeing the quality of the star ray search.

## 2 Related Work

*Optimization-based approaches:* Risk scores, which model $P(y = 1|\boldsymbol{x})$, are different from threshold classifiers, which predict either $y = 1$ or $y = -1$ given $\boldsymbol{x}$. Most work in the area of optimization of integer-valued sparse linear models focuses on classifiers, not risk scores [5, 6, 9, 32, 33, 37, 40, 46]. This difference is important, because a classifier generally cannot be calibrated well for use in risk scoring: only its single decision point is optimized. Despite this, several works use the hinge loss to calibrate predictions [6, 9, 32]. All of these optimization-based algorithms use mathematical programming solvers (i.e., integer programming solvers), which tend to be slow and cannot be used on larger problems. However, they can handle both feature selection and integer constraints.

To directly optimize risk scores, typically the logistic loss is used. The RiskSLIM algorithm [39] optimizes the logistic loss regularized with $\ell_0$ regularization, subject to integer constraints on the coefficients. RiskSLIM uses callbacks to a MIP solver, alternating between solving linear programs and using branch-and-cut to divide and reduce the search space. The branch-and-cut procedure needs to keep track of unsolved nodes, whose number increases exponentially with the size of the feature space. Thus, RiskSLIM's major challenge is scalability.

*Local search-based approaches:* As discussed earlier, a natural way to produce a scoring system or risk score is by selecting features manually and rounding logistic regression coefficients or hinge-loss solutions to integers [10, 11, 39]. While rounding is fast, rounding errors can cause the solution quality to be much worse than that of the optimization-based approaches. Several works have proposed improvements over traditional rounding. In Randomized Rounding [10], each coefficient is rounded up or down randomly, based on its continuous coefficient value. However, randomized rounding does not seem to perform well in practice. Chevaleyre [10] also proposed Greedy Rounding, where coefficients are rounded sequentially. While this technique aimed to provide theoretical guarantees for the hinge loss, we identified a serious flaw in the argument, rendering the bounds incorrect (see Appendix B). The RiskSLIM paper [39] proposed SequentialRounding, which, at each iteration, chooses a coefficient to round up or down, making the best choice according to the regularized logistic loss. This gives better solutions than other types of rounding, because the coefficients are considered together through their performance on the loss function, not independently.

A drawback of SequentialRounding is that it considers rounding up or down only to the nearest integer from the continuous solution. By considering *multipliers*, we consider a much larger space of possible solutions. The idea of multipliers (i.e., "scale and round") is used for medical scoring systems [11], though, as far as we know, it has been used only with traditional rounding rather than SequentialRounding, which could easily lead to poor performance, and we have seen no previous work that studies how to perform scale-and-round in a systematic, computationally efficient way. While the general idea of scale-and-round seems simple, it is not: there are an infinite number of possible multipliers, and, for each one, a number of possible nearby integer coefficient vectors that is the size of a hypercube, expanding exponentially in the search space.

*Sampling Methods:* The Bayesian method of Ertekin et al. [16] samples scoring systems, favoring those that are simpler and more accurate, according to a prior. "Pooling" [39] creates multiple models through sampling along the regularization path of ElasticNet. As discussed, when regularization is tuned high enough to induce sparse solutions, it results in substantial bias and low-quality solutions (see [37, 39] for numerous experiments on this point). Note that there is a literature on finding diverse solutions to mixed-integer optimization problems [e.g., 1], but it focuses only on linear objective functions.

**Algorithm 1** FasterRisk($\mathcal{D}$,$k$,$C$,$B$,$\epsilon$,$T$,$N_m$) $\rightarrow \{(\boldsymbol{w}^{+t}, w_0^{+t}, m_t)\}_t$

---

**Input:** dataset $\mathcal{D}$ (consisting of feature matrix $\boldsymbol{X} \in \mathbb{R}^{n \times p}$ and labels $\boldsymbol{y} \in \mathbb{R}^n$), sparsity constraint $k$, coefficient constraint $C = 5$, beam search size $B = 10$, tolerance level $\epsilon = 0.3$, number of attempts $T = 50$, number of multipliers to try $N_m = 20$.

**Output:** a pool $P$ of scoring systems $\{(\boldsymbol{w}^t, w_0^t), m^t\}$ where $t$ is the index enumerating all found scoring systems with $\|\boldsymbol{w}^t\|_0 \leq k$ and $\|\boldsymbol{w}^t\|_\infty \leq C$ and $m^t$ is the corresponding multiplier.

  1: Call Algorithm 2 SparseBeamLR($\mathcal{D}, k, C, B$) to find a high-quality solution $(\boldsymbol{w}^*, w_0^*)$ to the sparse logistic regression problem with continuous coefficients satisfying a box constraint, i.e., solve Problem (3). (Algorithm SparseBeamLR will call Algorithm ExpandSuppBy1 as a subroutine, which grows the solution by beam search.)
  2: Call Algorithm 5 CollectSparseDiversePool($(\boldsymbol{w}^*, w_0^*), \epsilon, T$), which solves Problem (4). Place its output $\{(\boldsymbol{w}^t, w_0^t)\}_t$ in pool $P = \{\boldsymbol{w}^*, w_0^*\}$. $P \leftarrow P \cup \{(\boldsymbol{w}^t, w_0^t)\}_t$.
  3: Send each member $t$ in the pool $P$, which is $(\boldsymbol{w}^t, w_0^t)$, to Algorithm 3 StarRaySearch $(\mathcal{D}, (\boldsymbol{w}^t, w_0^t), C, N_m)$ to perform a line search among possible multiplier values and obtain an integer solution $(\boldsymbol{w}^{+t}, w_0^{+t})$ with multiplier $m_t$. Algorithm 3 calls Algorithm 6 Auxiliary-LossRounding which conducts the rounding step.
  4: Return the collection of risk scores $\{(\boldsymbol{w}^{+t}, w_0^{+t}, m_t)\}_t$. If desired, return only the best model according to the logistic loss.

---

## 3  Methodology

Define dataset $\mathcal{D} = \{1, \boldsymbol{x}_i, y_i\}_{i=1}^n$ (1 is a static feature corresponding to the intercept) and scaled dataset as $\frac{1}{m} \times \mathcal{D} = \{\frac{1}{m}, \frac{1}{m}\boldsymbol{x}_i, y_i\}_{i=1}^n$, for a real-valued $m$. Our goal is to produce high-quality risk scores within a few minutes on a small personal computer. We start with an optimization problem similar to RiskSLIM's [39], which minimizes the logistic loss subject to sparsity constraints and integer coefficients:

$$\min_{\boldsymbol{w}, w_0} L(\boldsymbol{w}, w_0, \mathcal{D}), \quad \text{where } L(\boldsymbol{w}, w_0, \mathcal{D}) = \sum_{i=1}^n \log(1 + \exp(-y_i(\boldsymbol{x}_i^T \boldsymbol{w} + w_0))) \quad (1)$$

$$\text{such that} \quad \|\boldsymbol{w}\|_0 \leq k \text{ and } \boldsymbol{w} \in \mathbb{Z}^p, \ \forall j \in [1, .., p] \ w_j \in [-5, 5], \ w_0 \in \mathbb{Z}.$$

In practice, the range of these box constraints $[-5, 5]$ is user-defined and can be different for each coefficient. (We use 5 for ease of exposition.) The sparsity constraint $\|\boldsymbol{w}\|_0 \leq k$ or integer constraints $\boldsymbol{w} \in \mathbb{Z}^p$ make the problem NP-hard, and this is a difficult mixed-integer nonlinear program. Transforming the original features to all possible dummy variables, which is a standard type of preprocessing [e.g., 24], changes the model into a (flexible) generalized additive model; such models can be as accurate as the best machine learning models [39, 42]. Thus, we generally process variables in $\boldsymbol{x}$ to be binary.

To make the solution space substantially larger than $[-5, -4, ..., 4, 5]^p$, we use *multipliers*. The problem becomes:

$$\min_{\boldsymbol{w}, w_0, m} L\left(\boldsymbol{w}, w_0, \frac{1}{m}\mathcal{D}\right), \text{ where } L\left(\boldsymbol{w}, w_0, \frac{1}{m}\mathcal{D}\right) = \sum_{i=1}^n \log\left(1 + \exp\left(-y_i \frac{\boldsymbol{x}_i^T \boldsymbol{w} + w_0}{m}\right)\right) \quad (2)$$

$$\text{such that } \|\boldsymbol{w}\|_0 \leq k, \boldsymbol{w} \in \mathbb{Z}^p, \ \forall j \in [1, .., p], \ w_j \in [-5, 5], \ w_0 \in \mathbb{Z}, \ m > 0.$$

Note that the use of multipliers does not weaken the interpretability of the risk score: the user still sees integer risk scores composed of values $w_j \in \{-5, -4, .., 4, 5\}, w_0 \in \mathbb{Z}$. Only the risk conversion table is calculated differently, as $P(Y = 1|\boldsymbol{x}) = 1/(1 + e^{-f(\boldsymbol{x})})$ where $f(\boldsymbol{x}) = \frac{1}{m}(\boldsymbol{w}^T \boldsymbol{x} + w_0)$.

Our method proceeds in three steps, as outlined in Algorithm 1. In the first step, it approximately solves the following **sparse logistic regression** problem with a box constraint (but not integer constraints), detailed in Section 3.1 and Algorithm 2.

$$(\boldsymbol{w}^*, w_0^*) \in \underset{\boldsymbol{w}, w_0}{\operatorname{argmin}} L(\boldsymbol{w}, w_0, \mathcal{D}), \ \|\boldsymbol{w}\|_0 \leq k, \boldsymbol{w} \in \mathbb{R}^p, \forall j \in [1, ..., p], \ \boldsymbol{w}_j \in [-5, 5], w_0 \in \mathbb{R}.$$
$$(3)$$

The algorithm gives an accurate and sparse real-valued solution $(\boldsymbol{w}^*, w_0^*)$.

The second step produces **many near-optimal sparse logistic regression solutions**, again without integer constraints, detailed in Section 3.2 and Algorithm 5. Algorithm 5 uses $(\boldsymbol{w}^*, w_0^*)$ from the

first step to find a set $\{(\boldsymbol{w}^t, w_0^t)\}_t$ such that for all $t$ and a given threshold $\epsilon_{\boldsymbol{w}}$:

$$(\boldsymbol{w}^t, w_0^t) \text{ obeys } L(\boldsymbol{w}^t, w_0^t, \mathcal{D}) \leq L(\boldsymbol{w}^*, w_0^*, \mathcal{D}) \times (1 + \epsilon_{\boldsymbol{w}^*}) \tag{4}$$
$$\|\boldsymbol{w}^t\|_0 \leq k, \ \boldsymbol{w}^t \in \mathbb{R}^p, \ \forall j \in [1, ..., p], \ w_j^t \in [-5, 5], w_0^t \in \mathbb{R}.$$

After these steps, we have a pool of almost-optimal sparse logistic regression models. In the third step, for each coefficient vector in the pool, we **compute a risk score**. It is a feasible integer solution $(\boldsymbol{w}^{+t}, w_0^{+t})$ to the following, which includes a positive multiplier $m^t > 0$:

$$L\left(\boldsymbol{w}^{+t}, w_0^{+t}, \frac{1}{m^t}\mathcal{D}\right) \leq L(\boldsymbol{w}^t, w_0^t, \mathcal{D}) + \epsilon_t, \tag{5}$$
$$\boldsymbol{w}^{+t} \in \mathbb{Z}^p, \ \forall j \in [1, ..., p], w_j^{+t} \in [-5, 5], w_0^{+t} \in \mathbb{Z},$$

where we derive a tight theoretical upper bound on $\epsilon_t$. A detailed solution to (5) is shown in Algorithm 6 in Appendix A. We solve the optimization problem for a large range of multipliers in Algorithm 3 for each coefficient vector in the pool, choosing the best multiplier for each coefficient vector. This third step yields a large collection of risk scores, all of which are approximately as accurate as the best sparse logistic regression model that can be obtained. All steps in this process are fast and scalable.

---

**Algorithm 2** SparseBeamLR($\mathcal{D}, k, C, B$) $\rightarrow (\boldsymbol{w}, w_0)$

---

**Input:** dataset $\mathcal{D}$, sparsity constraint $k$, coefficient constraint $C$, and beam search size $B$.
**Output:** a sparse continuous coefficient vector $(\boldsymbol{w}, w_0)$ with $\|\boldsymbol{w}\|_0 \leq k, \|\boldsymbol{w}\|_\infty \leq C$.
1: Define $N_+$ and $N_-$ as numbers of positive and negative labels, respectively.
2: $w_0 \leftarrow \log(-N_+/N_-), \boldsymbol{w} \leftarrow \mathbf{0}$         ▷*Initialize the intercept and coefficients.*
3: $\mathcal{F} \leftarrow \emptyset$         ▷*Initialize the collection of found supports as an empty set*
4: $(\mathcal{W}, \mathcal{F}) \leftarrow \text{ExpandSuppBy1}(\mathcal{D}, (\boldsymbol{w}, w_0), \mathcal{F}, B)$.     ▷*Returns $\leq B$ models of support 1*
5: **for** $t = 2, ..., k$ **do**         ▷*Beam search to expand the support*
6:     $\mathcal{W}_{\text{tmp}} \leftarrow \emptyset$
7:     **for** $(\boldsymbol{w}', w_0') \in \mathcal{W}$ **do**         ▷*Each of these has support $t - 1$*
8:         $(\mathcal{W}', \mathcal{F}) \leftarrow \text{ExpandSuppBy1}(\mathcal{D}, (\boldsymbol{w}', w_0'), \mathcal{F}, B)$.   ▷*Returns $\leq B$ models with supp. $t$.*
9:         $\mathcal{W}_{\text{tmp}} \leftarrow \mathcal{W}_{\text{tmp}} \cup \mathcal{W}'$
10:     **end for**
11:     Reset $\mathcal{W}$ to be the $B$ solutions in $\mathcal{W}_{\text{tmp}}$ with the smallest logistic loss values.
12: **end for**
13: Pick $(\boldsymbol{w}, w_0)$ from $\mathcal{W}$ with the smallest logistic loss.
14: Return $(\boldsymbol{w}, w_0)$.

---

### 3.1 High-quality Sparse Continuous Solution

There are many different approaches for sparse logistic regression, including $\ell_1$ regularization [35], ElasticNet [48], $\ell_0$ regularization [13, 24], and orthogonal matching pursuit (OMP) [14, 25], but none of these approaches seem to be able to handle both the box constraints and the sparsity constraint in Problem 3, so we developed a new approach. This approach, in Algorithm 2, SparseBeamLR, uses beam search for best subset selection: each iteration contains several coordinate descent steps to determine whether a new variable should be added to the support, and it clips coefficients to the box $[-5, 5]$ as it proceeds. Hence the algorithm is able to determine, before committing to the new variable, whether it is likely to decrease the loss while obeying the box constraints. This beam search algorithm for solving (3) implicitly uses the assumption that one of the best models of size $k$ implicitly contains variables of one of the best models of size $k - 1$. This type of assumption has been studied in the sparse learning literature [14] (Theorem 5). However, we are not aware of any other work that applies box constraints or beam search for sparse logistic regression. In Appendix E, we show that our method produces better solutions than the OMP method presented in [14].

Algorithm 2 calls the ExpandSuppBy1 Algorithm, which has two major steps. The detailed algorithm can be found in Appendix A. For the first step, given a solution $\boldsymbol{w}$, we perform optimization on each single coordinate $j$ outside of the current support $supp(\boldsymbol{w})$:

$$d_j^* \in \underset{d \in [-5, 5]}{\arg\min} L(\boldsymbol{w} + d\boldsymbol{e}_j, w_0, \mathcal{D}) \text{ for } \forall j \text{ where } w_j = 0. \tag{6}$$

Vector $\boldsymbol{e}_j$ is 1 for the $j$th coordinate and 0 otherwise. We find $d_j^*$ for each $j$ through an iterative thresholding operation, which is done on all coordinates in parallel, iterating several ($\sim 10$) times:

$$\text{for iteration } i: d_j \leftarrow \text{Threshold}(j, d_j, \boldsymbol{w}, w_0, \mathcal{D}) := \min(\max(c_{d_j}, -5), 5), \qquad (7)$$

where $c_{d_j} = d_j - \frac{1}{l_j}\nabla_j L(\boldsymbol{w} + d_j\boldsymbol{e}_j, w_0, \mathcal{D})$, and $l_j$ is a Lipschitz constant on coordinate $j$ [24]. Importantly, we can perform Equation 7 on all $j$ where $w_j = 0$ in parallel using matrix form.

For the second step, after the parallel single coordinate optimization is done, we pick the top $B$ indices ($j$'s) with the smallest logistic losses $L(\boldsymbol{w} + d_j^*\boldsymbol{e}_j)$ and fine tune on the new support:

$$\boldsymbol{w}_{\text{new}}^j, w_{0\text{new}}^j \in \operatorname*{argmin}_{\boldsymbol{a} \in [-5,5]^p, b} L(\boldsymbol{a}, b, \mathcal{D}) \text{ with } supp(\boldsymbol{a}) = supp(\boldsymbol{w}) \cup \{j\}. \qquad (8)$$

This can be done again using a variant of Equation 7 iteratively on all the coordinates in the new support. We get $B$ pairs of $(\boldsymbol{w}_{\text{new}}^j, w_{0\text{new}}^j)$ through this ExpandSuppBy1 procedure, and the collection of these pairs form the set $\mathcal{W}'$ in Line 8 of Algorithm 2.

At the end, Algorithm 2 (SparseBeamLR) returns the best model with the smallest logistic loss found by the beam search procedure. This model satisfies both the sparsity and box constraints.

## 3.2 Collect Sparse Diverse Pool (Rashomon Set)

We now collect the sparse diverse pool. In Section 3.1, our goal was to find a sparse model $(\boldsymbol{w}^*, w_0^*)$ with the smallest logistic loss. For high dimensional features or in the presence of highly correlated features, there could exist many sparse models with almost equally good performance [7]. This set of models is also known as the Rashomon set. Let us find those and turn them into risk scores. We first predefine a tolerance gap level $\epsilon$ (hyperparameter, usually set to 0.3). Then, we delete a feature with index $j_-$ in the support $supp(\boldsymbol{w}^*)$ and add a new feature with index $j_+$. We select each new index to be $j_+$ whose logistic loss is within the tolerance gap:

$$\text{Find all } j_+ \text{ s.t. } \min_{a \in [-5,5]} L(\boldsymbol{w}^* - w_{j_-}^*\boldsymbol{e}_{j_-} + a\boldsymbol{e}_{j_+}, w_0, \mathcal{D}) \leq L(\boldsymbol{w}^*, w_0^*, \mathcal{D})(1 + \epsilon). \qquad (9)$$

We fine-tune the coefficients on each of the new supports and then save the new solution in our pool. Details can be found in Algorithm 5. Swapping one feature at a time is computationally efficient, and our experiments show it produces sufficiently diverse pools over many datasets. We call this method the CollectSparseDiversePool Algorithm.

## 3.3 "Star Ray" Search for Integer Solutions

The last challenge is how to get an integer solution from a continuous solution. To achieve this, we use a "star ray" search that searches along each "ray" of the star, extending each continuous solution outward from the origin using many values of a multiplier, as shown in Algorithm 3. The star ray search provides much more flexibility in finding a good integer solution than simple rounding. The largest multiplier $m_{\max}$ is set to $5/\max_j(|w_j^*|)$ which will take one of the coefficients to the boundary of the box constraint at 5. We set the smallest multiplier to be 1.0 and pick $N_m$ (usually 20) equally spaced points from $[m_{\min}, m_{\max}]$. If $m_{\max} = 1$, we set $m_{\min} = 0.5$ to allow shrinkage of the coefficients. We scale the coefficients and datasets with each multiplier and round the coefficients to integers using the sequential rounding technique in Algorithm 6. For each continuous solution (each "ray" of the "star"), we report the integer solution and multiplier with the smallest logistic loss. This process yields our collection of risk scores. Note here that a standard line search along the multiplier does not work, because the rounding error is highly non-convex.

We briefly discuss how the sequential rounding technique works. Details of this method can be found in Appendix A. We initialize $\boldsymbol{w}^+ = \boldsymbol{w}$. Then we round the fractional part of $\boldsymbol{w}^+$ one coordinate at a time. At each step, some of the $w_j^+$'s are integer-valued (so $w_j^+ - w_j$ is nonzero) and we pick the coordinate and rounding operation (either floor or ceil) based on which can minimize the following objective function, where we will round to an integer at coordinate $r^*$:

$$r^*, v^* \in \operatorname*{argmin}_{r,v} \sum_{i=1}^n l_i^2 \left( x_{ir}(v - w_r) + \sum_{j \neq r} x_{ij}(w_j^+ - w_j) \right)^2, \qquad (10)$$

$$\text{subject to } r \in \{j \mid w_j^+ \notin \mathbb{Z}\} \text{ and } v \in \{\lfloor w_r^+ \rfloor, \lceil w_r^+ \rceil\},$$

---

**Algorithm 3** StarRaySearch($\mathcal{D}, (\boldsymbol{w}, w_0), C, N_m) \rightarrow (\boldsymbol{w}^+, w_0^+), m$

---

**Input:** dataset $\mathcal{D}$, a sparse continuous solution $(\boldsymbol{w}, w_0)$, coefficient constraint $C$, and number of multipliers to try $N_m$.
**Output:** a sparse integer solution $(\boldsymbol{w}^+, w_0^+)$ with $\|\boldsymbol{w}^+\|_\infty \leq C$ and multiplier $m$.

1: Define $m_{\max} \leftarrow C/\max|\boldsymbol{w}|$ as discussed in Section 3.3. If $m_{\max} = 1$, set $m_{\min} \leftarrow 0.5$; if $m_{\max} > 1$, set $m_{\min} \leftarrow 1$.
2: Pick $N_m$ equally spaced multiplier values $m_l \in [m_{\min}, m_{\max}]$ for $l \in [1, ..., N_m]$ and call this set $\mathcal{M} = \{m_l\}_l$.
3: Use each multiplier to scale the good continuous solution $(\boldsymbol{w}, w_0)$, to obtain $(m_l\boldsymbol{w}, m_l w_0)$, which is a good continuous solution to the rescaled dataset $\frac{1}{m_l}\mathcal{D}$.
4: Send each rescaled solution $(m_l\boldsymbol{w}, m_l w_0)$ and its rescaled dataset $\frac{1}{m_l}\mathcal{D}$ to Algorithm 6 AuxiliaryLossRounding($\frac{1}{m_l}\mathcal{D}, m_l\boldsymbol{w}, m_l w_0$) for rounding. It returns $(\boldsymbol{w}^{+l}, w_0^{+l}, m_l)$, where $(\boldsymbol{w}^{+l}, w_0^{+l})$ is close to $(m_l\boldsymbol{w}, m_l w_0)$, and where $(\boldsymbol{w}^{+l}, w_0^{+l})$ on $\frac{1}{m_l}\mathcal{D}$ has a small logistic loss.
5: Evaluate the logistic loss to pick the best multiplier $l^* \in \operatorname{argmin}_l L(\boldsymbol{w}^{+l}, w_0^{+l}, \frac{1}{m^l}\mathcal{D})$
6: Return $(\boldsymbol{w}^{+l^*}, w_0^{+l^*})$ and $m_{l^*}$.

---

where $l_i$ is the Lipschitz constant restricted to the rounding interval and can be computed as $l_i = 1/(1 + \exp(y_i\boldsymbol{x}_i^T\boldsymbol{\gamma}_i))$ with $\gamma_{ij} = \lfloor w_j \rfloor$ if $y_i x_{ij} > 0$ and $\gamma_{ij} = \lceil w_j \rceil$ otherwise. (The Lipschitz constant here is much smaller than the one in Section 3.1 due to the interval restriction.) After we select $r^*$ and find value $v^*$, we update $\boldsymbol{w}^+$ by setting $w_{r^*}^+ = v^*$. We repeat this process until $\boldsymbol{w}^+$ is on the integer lattice: $\boldsymbol{w}^+ \in \mathbb{Z}^p$. The objective function in Equation 10 can be understood as an auxiliary upper bound of the logistic loss. Our algorithm provides an upper bound on the difference between the logistic losses of the continuous solution and the final rounded solution before we start the rounding algorithm (Theorem 3.1 below). Additionally, during the sequential rounding procedure, we do not need to perform expensive operations such as logarithms or exponentials as required by the logistic loss function; the bound and auxiliary function require only sums of squares, not logarithms or exponentials. Its derivation and proof are in Appendix C.

**Theorem 3.1.** *Let $\boldsymbol{w}$ be the real-valued coefficients for the logistic regression model with objective function $L(\boldsymbol{w}) = \sum_{i=1}^n \log(1 + \exp(-y_i\boldsymbol{x}_i^T\boldsymbol{w}))$ (the intercept is incorporated). Let $\boldsymbol{w}^+$ be the integer-valued coefficients returned by the AuxiliaryLossRounding method. Furthermore, let $u_j = w_j - \lfloor w_j \rfloor$. Let $l_i = 1/(1 + \exp(y_i\boldsymbol{x}_i^T\boldsymbol{\gamma}_i))$ with $\gamma_{ij} = \lfloor w_j \rfloor$ if $y_i x_{ij} > 0$ and $\gamma_{ij} = \lceil w_j \rceil$ otherwise. Then, we have an upper bound on the difference between the loss $L(\boldsymbol{w})$ and the loss $L(\boldsymbol{w}^+)$:*

$$L(\boldsymbol{w}^+) - L(\boldsymbol{w}) \leq \sqrt{n \sum_{i=1}^n \sum_{j=1}^p (l_i x_{ij})^2 u_j (1 - u_j)}. \tag{11}$$

**Note.** *Our method has a higher prediction capacity than RiskSLIM: its search space is much larger.* Compared to RiskSLIM, our use of the multiplier permits a number of solutions that grows exponentially in $k$ as we increase the multiplier. To see this, consider that for each support of $k$ features, since logistic loss is convex, it contains a hypersphere in coefficient space. The volume of that hypersphere is (as usual) $V = \frac{\pi^{k/2}}{\Gamma(\frac{k}{2}+1)}r^k$ where $r$ is the radius of the hypersphere. If we increase the multiplier to 2, the grid becomes finer by a factor of 2, which is equivalent to increasing the radius by a factor of 2. Thus, the volume increases by a factor of $2^k$. In general, for maximum multiplier $m$, the search space is increased by a factor of $m^k$ over RiskSLIM.

## 4 Experiments

We experimentally focus on two questions: (1) How good is FasterRisk's solution quality compared to baselines? (§4.1) (2) How fast is FasterRisk compared with the state-of-the-art? (§4.2) In the appendix, we address three more questions: (3) How much do the sparse beam search, diverse pools, and multipliers contribute to our solution quality? (E.4) (4) How well-calibrated are the models produced by FasterRisk? (E.9) (5) How sensitive is FasterRisk to each of the hyperparameters in the algorithm? (E.10)

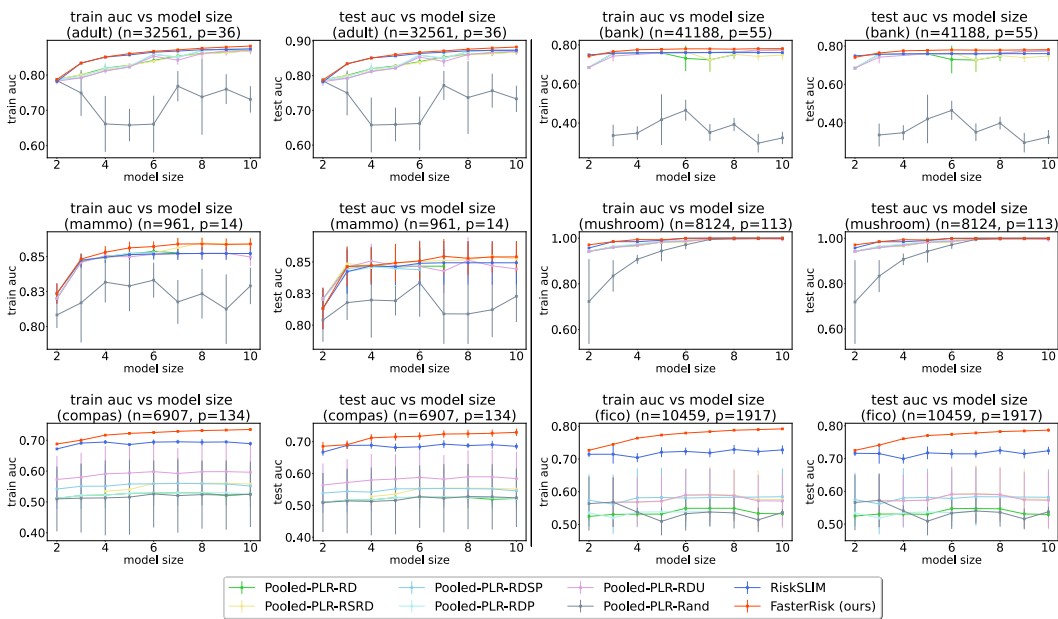

Figure 3: Performance comparison. FasterRisk outperforms all baselines due to its larger hypothesis space. On the datasets with highly-correlated variables such as COMPAS and FICO (both in the bottom row), FasterRisk outperforms other methods by a large margin.

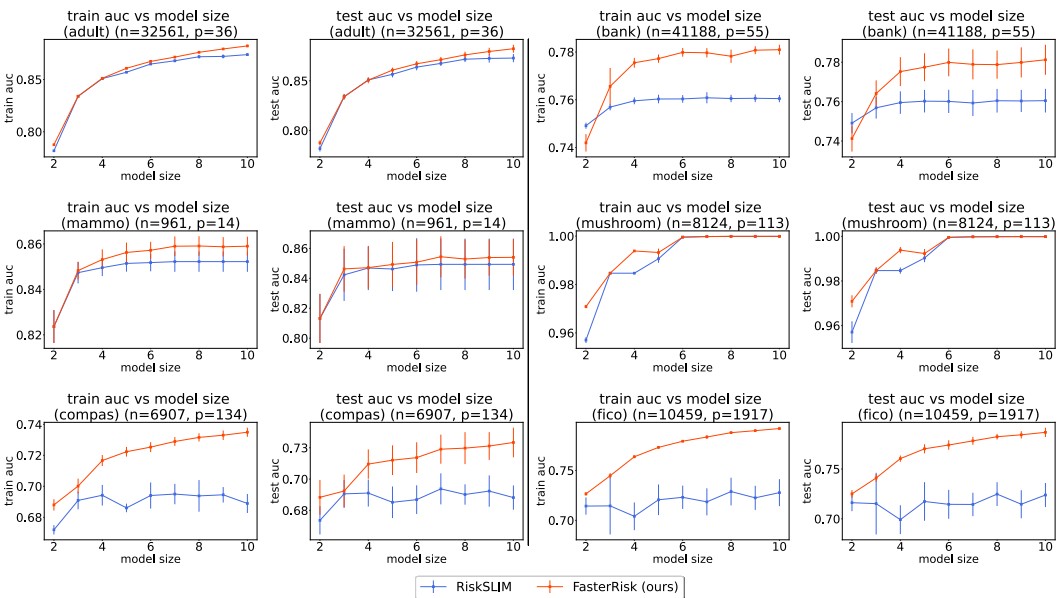

Figure 4: Performance comparison between FasterRisk and RiskSLIM.

We compare with RiskSLIM (the current state-of-the-art), as well as algorithms Pooled-PLR-RD, Pooled-PLR-RSRD, Pooled-PRL-RDSP, Pooled-PLR-Rand and Pooled-PRL-RDP. These algorithms were all previously shown to be inferior to RiskSLIM [39]. These methods first find a pool of sparse continuous solutions using different regularizations of ElasticNet (hence the name "Pooled Penalized Logistic Regression" – Pooled-PLR) and then round the coefficients with different techniques. Details are in Appendix D.3. The best solution is chosen from this pool of integer solutions that obeys the sparsity and box constraints and has the smallest logistic loss. We also compare with the baseline AutoScore [44]. However, on some datasets, the results produced by AutoScore are so poor that they distort the AUC scale, so we show those results only in Appendix E.11. As there is no publicly

available code for any of [10, 16, 32, 33], they do not appear in the experiments. For each dataset, we perform 5-fold cross validation and report training and test AUC. Appendix D presents details of the datasets, experimental setup, evaluation metrics, loss values, and computing platform/environment. More experimental results appear in Appendix E.

## 4.1 Solution Quality

We first evaluate FasterRisk's solution quality. Figure 3 shows the training and test AUC on six datasets (results for training loss appear in Appendix E). **FasterRisk (the red line) outperforms all baselines, consistently obtaining the highest AUC scores on both the training and test sets.** Notably, our method obtains better results than RiskSLIM, which uses a mathematical solver and is the current state-of-the-art method for scoring systems. This superior performance is due to the use of multipliers, which increases the complexity of the hypothesis space. Figure 4 provides a more detailed comparison between FasterRisk and RiskSLIM. One may wonder whether running RiskSLIM longer would make this MIP-based method comparable to our FasterRisk, since the current running time limit for RiskSLIM is only 15 minutes. We extended RiskSLIM's running time limit up to 1 hour and show the comparison in Appendix E.8; FasterRisk still outperforms RiskSLIM by a large margin.

FasterRisk performs significantly better than the other baselines for two reasons. First, the continuous sparse solutions produced by ElasticNet are low quality for very sparse models. Second, it is difficult to obtain an exact model size by controlling $\ell_1$ regularization. For example, Pooled-PLR-RD and Pooled-PLR-RDSP do not have results for model size 10 on the mammo datasets, because no such model size exists in the pooled solutions after rounding.

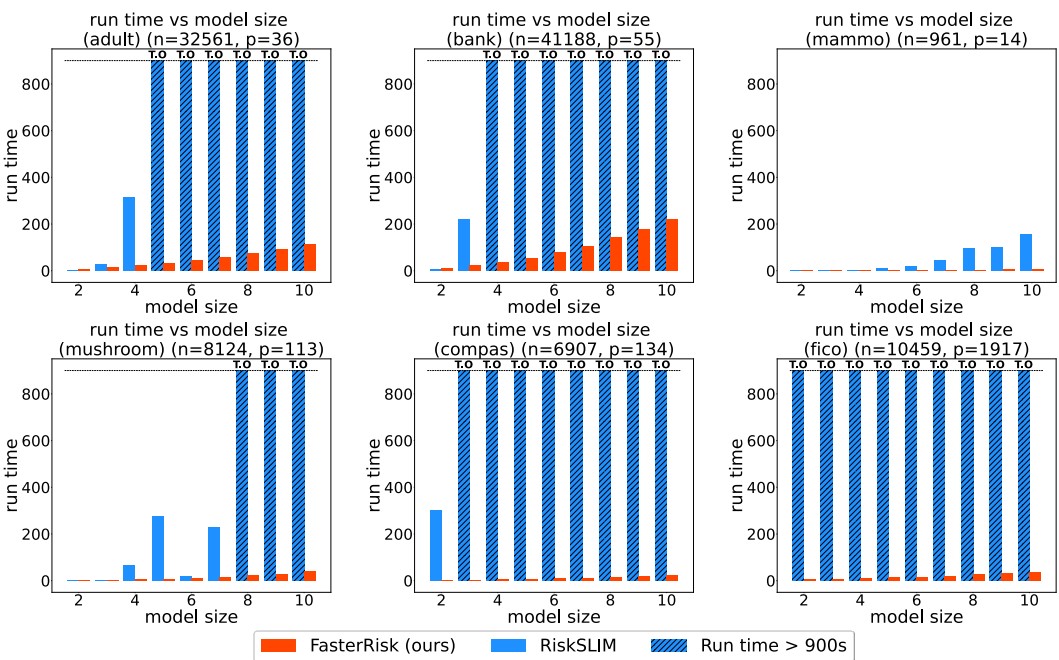

Figure 5: Runtime Comparison. Runtime (in seconds) versus model size for our method FasterRisk (in red) and the RiskSLIM (in blue). The shaded blue bars indicate cases that timed out (T.O.) at 900 seconds.

## 4.2 Runtime Comparison

The major drawback of RiskSLIM is its limited scalability. Runtime is important to allow interactive model development and to handle larger datasets. Figure 5 shows that **FasterRisk (red bars) is significantly faster than RiskSLIM (blue bars) in general**. We ran these experiments with a 900 second (15 minute) timeout. RiskSLIM finishes running on the small dataset mammo, but it times out on the larger datasets, timing out on models larger than 4 features for adult, larger than 3 features for bank, larger than 7 features for mushroom, larger than 2 features for COMPAS, and larger than 1

feature for FICO. RiskSLIM times out early on COMPAS and FICO datasets, suggesting that the MIP-based method struggles with high-dimensional and highly-correlated features. Thus, we see that FasterRisk tends to be both faster and more accurate than RiskSLIM.

## 4.3 Example Scoring Systems

The main benefit of risk scores is their interpretability. We place a few example risk scores in Table 1 to allow the reader to judge for themselves. More risk scores examples can be found in Appendix F.1. Additionally, we provide a pool of solutions for the top 12 models on the bank, mammo, and Netherlands datasets in Appendix F.2. Prediction performance is generally not the only criteria users consider when deciding to deploy a model. Provided with a pool of solutions that perform equally well, a user can choose the one that best incorporates domain knowledge [45]. After the pool of models is generated, interacting with the pool is essentially computationally instantaneous. Finally, we can reduce some models to relatively prime coefficients or transform some features for better interpretability. Examples of such transformations are given in Appendix G.1.

| 1. | no high school diploma | -4 points | | ... |
|---|---|---|---|---|
| 2. | high school diploma only | -2 points | + | ... |
| 3. | age 22 to 29 | -2 points | + | ... |
| 4. | any capital gains | 3 points | + | ... |
| 5. | married | 4 points | + | ... |
| | | **SCORE** | = | |

| **SCORE** | <-4 | -3 | -2 | -1 | 0 |
|---|---|---|---|---|---|
| **RISK** | <1.3% | 2.4% | 4.4% | 7.8% | 13.6% |
| **SCORE** | 1 | 2 | 3 | 4 | 7 |
| **RISK** | 22.5% | 35.0% | 50.5% | 65.0% | 92.2% |

(a) FasterRisk models for the adult dataset, predicting salary> 50K.

| 1. | odor=almond | -5 points | | ... |
|---|---|---|---|---|
| 2. | odor=anise | -5 points | + | ... |
| 3. | odor=none | -5 points | + | ... |
| 4. | odor=foul | 5 points | + | ... |
| 5. | gill size=broad | -3 points | + | ... |
| | | **SCORE** | = | |

| **SCORE** | -8 | -5 | -3 | ≥2 |
|---|---|---|---|---|
| **RISK** | 1.62% | 26.4% | 73.6% | >99.8% |

(b) FasterRisk model for the mushroom dataset, predicting whether a mushroom is poisonous.

Table 1: Example FasterRisk models

## 5 Conclusion

FasterRisk produces a collection of high-quality risk scores within minutes. Its performance owes to three key ideas: a new algorithm for sparsity- and box-constrained continuous models, using a pool of diverse solutions, and the use of the star ray search, which leverages multipliers and a new sequential rounding technique. FasterRisk is suitable for high-stakes decisions, and permits domain experts a collection of interpretable models to choose from.

## Code Availability

Implementations of FasterRisk discussed in this paper are available at `https://github.com/jiachangliu/FasterRisk`.

## Acknowledgements

The authors acknowledge funding from the National Science Foundation under grants IIS-2147061 and IIS-2130250, National Institute on Drug Abuse under grant R01 DA054994, Department of Energy under grants DE-SC0021358 and DE-SC0023194, and National Research Traineeship Program under NSF grants DGE-2022040 and CCF-1934964. We acknowledge the support of the Natural Sciences and Engineering Research Council of Canada (NSERC). Nous remercions le Conseil de recherches en sciences naturelles et en génie du Canada (CRSNG) de son soutien.

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
