# OpenReview forum: "FasterRisk: Fast and Accurate Interpretable Risk Scores"
_NeurIPS.cc/2022/Conference — NeurIPS 2022 Accept_

### Official Review · Reviewer_DdPk · 2022-07-06

**Rating:** 7
**Confidence:** 4
**Soundness:** 4 excellent
**Presentation:** 3 good
**Contribution:** 3 good

**Summary:**

The paper is focused on risk scores learning which are simple but efficient (in terms of performance) models. The main idea is to produce a pool of almost-optimal sparse continuous solutions with different support sets using a beam-search algorithm. Each of these solutions is explored: the real-values models are transformed into feasible integer-valued solutions along multipliers (what allows for a large space of possible solutions). The method is computationally efficient.

**Questions:**

It seems that m is defined quite late in text, and it is not clear from the beginning of Section 3 that it is the multiplier. Why did you decide to divide by m and not to multiply (if it is a multiplier)?

In Section 3.2. it is mentioned that "swapping one feature at a time is computationally efficient". I would say rather not efficient, if there are a lot of features.

I am not sure whether Section 4.3. is necessary. It underlines that there are not any results on real scores in the current submission. I guess the example provided in the Introduction (and Appendix) is enough.

I am curious to know what the range of m (multiplier) values was in your experiments?
How many intermediate (pool) models did you generate? And what is the percentage of reasonable final (integer) models?


**Limitations:**

The authors provide the limitations on page 3.

**Strengths And Weaknesses:**

Strengths. The paper clearly describes a novel three step framework to learn simple interpretable models. The numerical results are convincing.

Weaknesses. The method has three separate steps what can lead to some inconveniences (some kind of error cumulation is possible; coordinate descent can be long as well as the line search).
As also mentioned by the authors, real scores are not considered in the current contribution.

---

> ### Author Response · Authors · 2022-08-02
> **Response to Reviewer DdPk; All Requested New Experiments Are in Appendix G**
>
> 1. $\textbf{Are real scores considered in the current contribution?}$ By real scores, do you mean continuous coefficients? We want integer coefficients. Creating sparse continuous solutions as in Section 3.1 and 3.2 are intermediate milestones towards this goal. So we do this too.
>
> 2. $\textbf{Are there any possible large error accumulation?}$ Not according to our experiments. We don't lose error at any step so error doesn't accumulate.
>
> 3. $\textbf{How to understand the multiplier $m$?}$ When we do StarRaySearch in Algorithm 3, we shrink the feature matrix by $m$ and $\textit{multiply}$ the coefficients by $m$ (therefore the name ''multiplier''). When we calculate the risk probability, because the scores are based on the original features instead of shrunk features, we have to divide the total score by this multiplier $m$ to get the right probability.
>
> 4. $\textbf{Can you comment on swapping one feature at a time is computationally efficient?}$ Actually, swapping one feature for another is efficient. If you consider swapping 2 features, there are ${k \choose 2}$ choices, where $k$ is the number of nonzero features. 3 features requires you to choose among $\binom{k}{3}$ features. It is $\textit{much}$ computationally easier just to enumerate $k$ options for swapping.
>
> 5. $\textbf{What is the range of multiplier?}$ This has already been specified in Line 2 in Algorithm 3. We re-state here in words: if all coefficients have magnitude all less than 5, we want to stretch the coefficients with a multiplier. The largest value for the multiplier is $m_{max} = 5/max |w|$ because we want the coefficients to stay within the box constraints. The smallest multiplier is $m_{min}=1$. If some coefficients are already on the boundary of the box constraints (either +5 or -5), we explore shrinking the coefficients by a multiplier. We choose the smallest multiplier to be $m_{min}=0.5$ and the largest multiplier to be $m_{max}=1$. We pick $N_m$ (default value is 20) equally spaced multiplier values from the interval $[m_{min}, m_{max}]$. We have done perturbation study on this hyperparameter $N_m$ during rebuttal. If you are interested, please go to Appendix G4.4 to see the results.
>
> 6. $\textbf{How many intermediate pool models did you generate?}$ At most 50. See Appendix D4: Hyperparameter Specification.
>
> Thank you once again for your review!

---

### Official Review · Reviewer_aCoP · 2022-07-11

**Rating:** 7
**Confidence:** 5
**Soundness:** 3 good
**Presentation:** 3 good
**Contribution:** 3 good

**Summary:**

This paper aims to provide a fast algorithm to derive sparse risk scores that scales to high-dimensional datasets. The authors identified a few major limitations in current methods, and described how these were addressed by the three components in their proposed algorithm. In several experiments with low and high dimension data, the authors showed that their method outperformed the current state-of-the-art and several other baseline methods. The algorithm is implemented in stand-alone Python code, which is advantageous over competitors that rely on mathematical programming solvers.

**Questions:**

1. The authors focused on developing scores by finding integer sparse solutions, and showed some advantages of FasterRisk over the state-of-the-art. But I find the discussion of alternative approaches inadequate, therefore I could not fully appreciate the contribution of this work. For example, the authors pointed out two major limitations of building scores by rounding logistic regression coefficients: (i) $l_1$ and $l_0$ regularizations not able to get sparse solutions, and (ii) rounding of coefficients worsens performance by making the scores too coarse. But (i) may be resolved by using alternative variable selection methods and (ii) by using smallest non-zero coefficient to scale all coefficients and then rounding to larger integer values (e.g., total score ranging from 0 to 100). For example, a 2020 paper (https://doi.org/10.2196/21798) describes such an alternative approach that worked reasonably well in several clinical applications, and by separating variable selection from score development, domain experts are more easily engaged in the development process to ensure clinical meaningfulness and fairness. I find this paper lacking in discussion on this general approach. Could the authors include such more recent works in their discussion and method evaluation?

2. The authors stated in appendix that the choice of hyperparameters does not have much impact on performance, but did not provide empirical evidence. I am particularly concerned with the choice of tolerance gap level $\{epsilon}=0.3$ (equation (9)), meaning we are willing to tolerate up to 30% increase in loss when expanding to “equally good” scores. Without detailed explanation, 30% seems too large to me. Can the authors justify their choice empirically or by citing related literature? I would also like to see some empirical results regarding change in other hyperparameters.

3. Although the authors generated a pool of “equally good” scores, they did not seem to make use of them other than selecting the best-performing one to report. This pool of scores could be useful for users to select well-performing AND fair scores. This is related to Limitations below.

**Limitations:**

Fairness of scores developed from the proposed algorithm is not adequately discussed. Related to Question 3 above, a naïve application of the proposed method may lead to unfair risk scores. For example, in Table 3 of Appendix F, the method generated a 3-variable risk score to predict salary>30K using education level and marital status, and this direct link of being married with salary level is highly debatable. Marital status might represent a mixed effect of age and socio-economic status, and it may be better to use the latter in the score for more meaningful interpretation. Since the authors have generated a pool of “equally good” scores that make use of alternative predictors in a step of the algorithm, I suggest the authors make use of this pool to help users balance performance and fairness.

**Strengths And Weaknesses:**

This paper describes the authors’ original work to resolve several methodological difficulties in current development of risk scores. The writing is clear in defining several major challenges the authors aimed to address, and the proposed algorithm consists of separate components to address them. In addition to evaluating the algorithm with respect to baselines, the authors also showed the importance of each component by assessing the reduction in performance without them.

My major concern is the scope of this study, which affects its quality and significance. The authors aim to develop a fast and well-performing algorithm, FasterRisk, to develop sparse risk scores, which, if successful, would be very useful to healthcare applications. But when discussing related work, the authors did not include some recent works that have partially addressed some of the limitations the authors proposed to address (elaborated in Question 1 below). When evaluating FasterRisk, there is practically only one competitor algorithm, and FasterRisk only had marginal advantage in performance (in most experiments). FasterRisk is indeed much faster than the competitor, but by timing out the run time at 15 minutes, I am not convinced that the competitor is slow enough to be concerning in practice.

The clarity in mathematics notations can be improved. The equations became difficult to follow when the author used some notations without introducing them. For example, $\{epsilon}_w$ in equation (4) and $\{epsilon}_t$ in equation (5) lack bounds, and it is difficult to understand what the arbitrary a, b, c, d, e in equations (6)-(9) stand for. These affect my trust in the work.

---

> ### Author Response · Authors · 2022-08-02
> **Response to Reviewer aCoP; All Requested New Experiments Are in Appendix G**
>
> 1. $\textbf{Is the competitor RiskSLIM being slow concerning in practical applications?}$ Yes! Speed is very important in these settings: (1) When we cannot compute the answer at all using the slow method because it does not scale to reasonably-sized datasets. It could take over a week to compute the solution for even reasonably small datasets. (2) Interactive machine learning. Machine learning in the wild is essentially never a single run of an algorithm. Often times, the users want to explore the data and adjust various constraints along the way as they get more familiar with possible models. Fast speed allows users to go through this iteration process many times without interruptions (of several days perhaps) for the algorithm to run. This is where FasterRisk will be very useful in high stakes offline settings. This is because after the pool of models is generated within 5 minutes, interacting with the pool is essentially instantaneous, allowing users to interact with it. Please see  Appendix G1 (in the new appendix for reviewers, Appendix G).
>
> 2. $\textbf{Are there datasets where our method FasterRisk has significant performance (solution quality) advantage?}$ Yes! Please see Figure 5 (Appendix E1) and Figure 6 (Appendix E2). These three datasets have high dimensional features and feature correlation is also very high (See Table 2 in Appendix D1 for data set information), which make them challenging. Our method FasterRisk significantly outperforms other baselines in AUC. Additionally, please look at Figure 11 in Appendix E5 and Figures 18-21 in Appendix G1 for the logistic loss curves on the training set. FasterRisk achieves much lower logistic loss than RiskSLIM.
>
> 3. $\textbf{Can you clarify some mathematical notations?}$
>
>     3.1. $\epsilon_{w^*}$ in Eq 4 is a user-defined value for the gap tolerance level corresponding to the optimal solution $w^*$. If we set $\epsilon_{w^*}$ too close to 0, we may not find any solutions (since it is NP-hard to find the optimal solution); if we set $\epsilon_{w^*}$ too large, we could find too many solutions to evaluate and some solutions are not as good quality.
>
>     3.2. $\epsilon_{t}$ in Eq 5 is any arbitrary value that makes Eq 5 hold. We can think of $\epsilon_{t}$ as the loss difference between the rounded integer solution $\{w^{+t}, w_0^{+t}\}$ and original continuous solution $\{w^{t}, w_0^{t}\}$.
>
>     3.3. In Eq 6, $d$ is the coefficient on coordinate $j$. $\textbf{e}$${}_j$ is a unit vector with 1 on coordinate $j$ and 0 on other coordinates.
>
>     3.4. In Eq 7, $c_{d_j}$ is already defined in Line 182. It is the coefficient on coordiante $j$ after one step of coordinate descent.
>
>     3.5. In Eq 8, you can think of $a$ and $b$ as coefficients and intercept. We try to avoid using $w, w_0$ here because they appear on the left.
>
>     3.6. In Eq 9, you can think of $a$ as the coefficient on coordinate $j$.
>
> 4. $\textbf{Can we compare with AutoScore?}$ Thank you for providing this paper. We have cited this in Line 394. Please see Figure 25-27 in Appendix G3 for the comparison between AutoScore, RiskSLIM, and FasterRisk. FasterRisk outperforms AutoScore in all cases.
>
>     We are solving a more challenging optimization problem than AutoScore because AutoScore does not impose the box constraints on the coefficients during feature selection. We want scores to be small (in one digit) so that ordinary people can add/subtract numbers in their heads.
>
>     The philosophy of these two approaches is different. AutoScore is designed to involve the user in the optimization process. In contrast, FasterRisk solves the optimization automatically and engages the user only in selecting the best model. Our method FasterRisk is complementary to AutoScore's random forest-based feature selection approach; one can apply FasterRisk for feature selection by using our beam search method. Again, please see Figure 25-27 in Appendix G3 for the comparison between AutoScore, RiskSLIM, and FasterRisk.
>
> 5. $\textbf{Can we show results for other $\epsilon$ values for the diverse pool?}$ Please see Figure 31-33 in Appendix G4.2.
>
> 6. $\textbf{Can we show other empirical results regarding change in other hyperparameters?}$ In Figure 28-30 in Appendix G4.1, we changed the beam size hyperparameter; in Figure 34-36 in Appendix G4.3, we changed the number of attempts hyperparameter; in Figure 37-39 in Appendix G4.4, we changed the number of multipliers hyperparameter.
>
> 7. $\textbf{Can we show examples from the pool of solutions?}$ Yes! Please see Tables 30-41 in Appendix G5 for 12 models from the pool.
>
> 8. $\textbf{Add to the Checklist. Is there insufficient discussion of fairness?}$ Good point. We added a discussion in the checklist to say that even if a model is interpretable, it could still have negative societal bias, and looking at a variety of models from the pool could help find models that are more fair.
>
> Thank you so much for your review!

---

> > ### Comment · Reviewer_aCoP · 2022-08-10
> > **Thank you for addressing my concerns and questions**
> >
> > I believe this work is a valuable addition to the literature of scoring systems.
> >
> > One minor suggestion: In the response, the authors mentioned that "FasterRisk solves the optimization automatically and engages the user only in selecting the best model." While it is important to automate the learning process, it is also crucial to note that some automatically selected variables may not make sense to clinicians. Ideally, domain knowledge should be integrated into the scoring system as early as possible. Moreover, the "best model" might not be the most suitable model/solution in clinical practice. While FasterRisk has good potential in clinical applications, the authors are suggested to expand their discussions a little bit to reflect the importance of domain knowledge and actual clinical needs in implementations.

---

### Official Review · Reviewer_ifPK · 2022-07-11

**Rating:** 7
**Confidence:** 3
**Soundness:** 4 excellent
**Presentation:** 3 good
**Contribution:** 3 good

**Summary:**

This study proposed a novel method to accurately and efficiently generate a collection of high-quality risk scores based on the integration of the beam-search-based algorithm for LR, the generation of diverse high-quality solutions with different support sets, and the star search for integer solutions. It achieved SOTA performance with less time cost in some datasets.

**Questions:**


None

**Strengths And Weaknesses:**

Strengths:
1.	The proposed three-step framework includes a beam-search-based algorithm for logistic regression with box constraints and L0 regularization, the search algorithm to collect the sparse diverse pool with different support set, and the star search technique using multipliers, and a theorem guaranteeing the quality of the star search results. The whole methodology was solid and efficient.
2.	The introductions of the research context and related work were well-organized and clear.
3.	The proposed method achieved the SOTA performance with significantly less time (as shown in figure 4), showing its reliability and efficiency.
4.	Their theoretical discussion and supplement material were abundant, Moreover, they also conducted extensive experiments on performance, including performance comparison, efficiency, and ablation experiments.

Weaknesses:
1.	The examples of scoring systems in the Introduction seem out of date, there are many newer and recognized clinical scoring systems. It also should briefly introduce the traditional framework of the scoring system and its difference in methodology and performance with the proposed method.
2.	As shown in figure 3, the performance improvement of proposed methods seems not so significant, the biggest improvement in the bank dataset was ~0.02. Additionally, using some tables to directly show the key improvements may be more intuitive and detailed.
3.	Although extensive experiments and discussion on performance, in my opinion, its most significant improvement would be efficiency, and there are few discussions or ablation experiments on efficiency.
4.	The model AUC can assess the model discriminant ability, i.e., the probability of a positive case is bigger than that of a negative case, but may be hard to show its consistency between predicted score and actual risk. However, this consistency may be more crucial to the clinical scoring system (differentiated with classification task). Therefore, the related studies are encouraged to conduct calibration curves to show the agreement. It would be better to prove the feasibility of the generated scoring system?  The difference between the traditional method and our method can also be discussed in this paper.

---

> ### Author Response · Authors · 2022-08-02
> **Response to Reviewer ifPK; All Requested New Experiments Are in Appendix G**
>
> 1. $\textbf{Examples of scoring systems in the Introduction are out of date.}$ Indeed, we used a lot of historical examples including many established and popular medical scoring systems. We added a few more recent ones. See Line 42 for new citations, including several scoring systems for COVID-19 patients.
>
> 2. $\textbf{Can we discuss differences with traditional framework of the scoring system?}$ We have discussed this in Section 2 Related Work.
>
> 3. $\textbf{Can we provide more discussion/experiments on efficiency?}$ Yes! Please see the experiments and discussions in the new Appendix G1.
>
> 4. $\textbf{Is performance improvement significant?}$ The timing improvement is orders of magnitude smaller, which is the most important. RiskSLIM is guaranteed to produce optimal solutions (with respect to its search space) eventually, so we don't expect to always achieve a performance improvement with respect to it. However, since our search space is larger, we often see a performance improvement that is significant, especially on the three extra datasets in the Appendix. Please see Figure 5 (Appendix E1) and Figure 6 (Appendix E2). Additionally, please look at Figure 11 in Appendix E5 and Figures 18-21 in Appendix G1 for the logistic loss curves on the training set. FasterRisk achieves better logistic losses than RiskSLIM.
>
> 5. $\textbf{Can we provide calibration curves?}$ Yes! Please see Figure 22-24 in Appendix G2.
>
> 6. $\textbf{Can we prove the feasibility of the generated scoring system?}$ All scoring systems satisfy the constraints and are thus feasible. Sorry, perhaps we don't understand this question. Do you mind elaborating or paraphrasing the question?
>
> Thank you so much for your review!

---

### Official Review · Reviewer_ntN1 · 2022-07-12

**Rating:** 6
**Confidence:** 3
**Soundness:** 3 good
**Presentation:** 3 good
**Contribution:** 3 good

**Summary:**

The authors propose a method for efficiently automatically generating a pool of “risk scores” (sparse linear models with integer coefficients), involving (1) a beam search algorithm to identify a sparse set of features, (2) given the original set of features, identify a pool of sparse solutions with similar performance (but “diverse” set of features), and (3) “star ray” search to choose integer coefficients. They evaluate both the speed and accuracy of their approach on multiple benchmark datasets.

**Questions:**

- You mention  (line 54)  “We need an approach that exhibits the best of both worlds: speed fast enough to operate in a few  minutes on a laptop and optimization and search capability as powerful as that of the mathematical programming tools. Our method, FasterRisk, lies at this intersection.” — Of course, it makes sense to have a goal of developing an accurate model, but I’m wondering why it is important that risk scores must operate in a few minutes on a laptop. If these are risk scores for a high-impact situation, why can’t we run an analysis for an hour? Or even a week?

- Figure 4:   I find this figure a bit frustrating, because it’s not really allowing me to see how the baseline method scales compared to yours.  A 15 minute time-out seems kind of silly/arbitrary (in the real world, I’d expect people to be willing to train their methods for quite a while if it’s for a high stakes setting), and I would highly recommend allowing your baselines to run for longer (at least several hours) to show how the times actually scale (and then possibly display with a log scale as needed). It would also be helpful to share the number of rows and columns in each of the datasets to give a sense of scale.

- Section 4.3 Example scoring systems: how were these examples selected?   Also, you mention that risk scores offer interpretability – is this something that you argue is unique to your approach and not your baselines?

- Your algorithm is often described to return a “pool” of solutions, but I didn’t see much discussion of what  that pool actually looks like. For example, I would want to see how many solutions were produced, how different they are from each other, etc.   In a use case, would you expect the user to look through all of them and then choose one, or just defer to the lowest-error one?


**Limitations:**

The authors describe some limitations related to their algorithms, which is appreciated. However, they describe the potential negative societal impacts as “[N/A]” which seems like a huge oversight to me. As they say, “[risk scores] are possibly the most popular form of predictive model for high stakes decisions through the last century and are the standard form of model used in criminal justice  and medicine,” it seems obvious that any contribution they make to this field could have serious societal impacts, for better or worse.

Here’s an example of how this approach could be used problematically:  Let’s say we have a criminal justice scenario in which we have access to race and some other features that are essentially proxies for race (e.g., zip code). 	Now let’s assume a user is given a pool of “diverse” solutions by the FasterRisk approach, and they know that they don’t want a model that’s “racist”. They may notice one risk score has a highest accuracy but relies on race, so they decide they shouldn’t use that model. They then notice an almost identical model that has all the same features except race has been replaced by zip code, and they choose this model instead and deploy it in some real world scenario (e.g., recommending whether someone should be placed on parole).


**Strengths And Weaknesses:**

The authors describe an interesting method for quickly identifying risk scores in a diverse range of settings. The main improvement seems to be speed (which wasn’t very well quantified with respect to baselines), since performance-wise  it was similar to a previous approach – and I wonder how useful this speed would actually be in this high-stakes offline setting.

Originality/clarity: this seems to be a creative combination of past algorithms, and they described their algorithms in detail

Quality/significance: the authors evaluated their methods and baselines along both accuracy and speed metrics, and also shared extensive additional experiments in their supplement. One aspect that seemed problematic was that for speed plots, they cut off algorithms after 15 minutes, and it’s unclear exactly how their baselines scale because they tend to be censored after just one or two points along the x-axis. It would also be helpful to see if quantitatively, there are any significant differences in AUC between FasterRisk and alternative methods. Based on these two results, it would be easier to assess whether their method provides a meaningful contribution to real-world use cases of risk scores.

The authors also do not describe any considerations of societal impact which is an important factor given their suggested use cases.

---

> ### Author Response · Authors · 2022-08-02
> **Response to Reviewer ntN1; All Requested New Experiments Are in Appendix G**
>
> 1. $\textbf{Why is speed improvement important?}$ Speed is very important in these settings: (1) when we cannot compute the answer at all using the slow method because it does not scale to reasonably-sized datasets. It could take over a week to compute the solution for even reasonably small datasets. (2) Interactive design of models. Machine learning in the wild is generally never a single run of an algorithm. Often times, the users want to explore the data and adjust various constraints along the way as they get more familiar with possible models. Fast speed allows users to go through this iteration process many times without interruptions (of several days perhaps) for the algorithm to run. This is where FasterRisk will be very useful in high stakes offline settings. This is because after the pool of models is generated within 5 minutes, interacting with the pool is essentially instantaneous, allowing users to interact with it.
>
> 2. $\textbf{Are there datasets where there are some significant differences (beyond 2 percent) in AUC between FasterRisk and alternative methods?}$ Please see Figure 5 (Appendix E1) and Figure 6 (Appendix E2). These three datasets have high dimensional features and feature correlation is also very high (See Table 2 in Appendix D1 for data set information), which makes them challenging. Our method FasterRisk significantly outperforms other baselines in AUC. Additionally, please look at Figure 11 in Appendix E5 and Figures 18-21 in Appendix G1 for the logistic loss curves on the training set. FasterRisk is doing a much better optimization job than RiskSLIM because it can use a larger search space.
>
> 3. $\textbf{What happens when we run the baseline RiskSlim longer?}$ We added an experiment in Figures 18-21 in Appendix G1.1 and G1.2. When we run RiskSLIM for 1 hour, FasterRisk still outperforms RiskSLIM; when we run RiskSLIM longer for 4 days, FasterRisk outperforms RiskSLIM on 7 out of 9 datasets except on Mushroom and Spambase. (In the two remaining cases, the results are essentially tied.)
>
> 4. $\textbf{Can we share the number of rows and columns in each data set?}$ Yes, the dataset information is already shown in Table 2 of Appendix D1.
>
> 5. $\textbf{How were scoring system examples selected in section 4.3?}$ They were selected from the diverse pool based on the smallest logistic loss on the training set. It is in exactly the same manner as detailed in Line 253-254 for the pooled-PLR baselines.
>
> 6. $\textbf{Is interpretability unique to our approach?}$ No, all methods that produce risk scores are interpretable. This includes our approach and the baselines we compared with.
>
> 7. $\textbf{Can I see pool of solutions?}$ Sure! We have included several large tables in Appendix G5 in Table 30-41 of 12 models from the pool.
>
> 8. $\textbf{Do we expect users to look through all solutions in a pool or only the best one with the lowest error?}$ Ideally a user-interface would help guide the user from the lowest-error model to one that would best suit their needs. Error is generally not the only criteria users would consider when deciding to implement a model. Our contribution here is just to design the algorithm for finding these models, but we plan to develop this interface next. The user could also simply rank the models from the smallest to the largest error and choose one.
>
> 9. $\textbf{Why do we answer ``N/A'' for the checklist question about any negative societal impacts?}$ The reviewer makes an excellent point. We have edited this in the checklist to say that even if a model is interpretable, it can still have negative societal bias (though it is easier to check for such biases with scoring systems), and looking at a variety of models from the pool could help find models that are more fair. We placed this information into Appendix G, which is the new appendix for reviewers.
>
> Thank you so much for your review!

---

> > ### Comment · Reviewer_ntN1 · 2022-08-08
> > **Re: Response to Reviewer ntN1**
> >
> > Thanks for taking the time to thoughtfully answer my questions and address some of my concerns!
> >
> > I have some remaining concerns, and perhaps there have been some misunderstandings in terms of my original questions.  One of my main questions was about Figure 4, and how it was cut off at 15 minutes, so it's difficult to interpret how the runtime of RiskSLIM compares with FasterRisk. While it's nice to see a newer version of Figure 3 in Appendix G1.1 and G1.2 with a longer runtime, my concern about the interpretability of Figure 4 remains, and in the final version, it would be great to see a version of Figure 4 with a much higher y-axis limit.
> >
> > While I appreciate you answering the questions below, I'd like to see that some of these answers are actually incorporated into a published version of your paper. In particular, for the final version, I'd love to see revisions based on your comments:
> > - 1: why speed is important
> > - 4: # rows/columns  in datasets incorporated into the **MAIN** text, e.g. as part of figure 4, since readers shouldn't need to dig around in your appendix for this basic information
> > - 6: a comment that interpretability is a general feature of risk scores and not just your method
> > - 8: why it's useful to have a pool of solutions, and how you would expect them to be used in practice
> > - 9: ethical considerations (possibly in the discussion section, this could perhaps be combined with 6 and 8)

---

> > > ### Author Response · Authors · 2022-08-09
> > > **Response to Reviewer ntN1; Newly Requested Time Plot Is in Appendix G.7**
> > >
> > > We thank Reviewer ntN1 for the comments.
> > >
> > > Yes, *all* comments from *all* reviewers will be incorporated into the main manuscript. We haven’t done it yet for two reasons: (1) we didn’t have enough time since we were asked to perform a huge number of experiments during the short rebuttal phase and we didn't have enough time to crush it all carefully into NeurIPS' 9 page limit, and (2) we thought it would be helpful for reviewers if we kept the original main paper and appendix while providing only a new Section G titled ''Reviewer-requested Extra Experiments and Discussion'' (Page 43-69). The reason for this arrangement is that we thought the reviewers might want to compare the originally submitted results and newly requested results in Section G.
> > >
> > > With regards to the remaining concern on running time in the last reply, please see a new plot in Figure 40 under Appendix G.7. We think the replacement of Figure 4 with the new ones we propose in the appendix should do it. Just to recap: we have included new time results (time limit is 1h) in Figure 40 in Section G.7. Figure 4 and 7 in the original submission already shows that FasterRisk can finish running at most 5 minutes while RiskSLIM runs on most folds and datasets for 15 minutes (which was the original time limit) without finishing. In Figure 40 (page 70 in the appendix), we raised the time limit for RiskSLIM up to 1 hour. Most of the folds and datasets still do not finish. Therefore, using the 1-hour results of RiskSLIM show that FasterRisk has an even more impressive speed advantage. We also discussed in our earlier rebuttal about the necessity of faster run times - interactions with humans should be able to be done in real-time, not waiting over an hour between runs.
> > >
> > > We believe all your technical comments were addressed in our rebuttal and in the new Appendix G for reviewers. Again, we will definitely incorporate all your revision comments in the final version if this submission gets accepted. We understand that you may want the revision writing and plotting to be done now, but this new comment is posted on August 8, and the deadline for author-reviewer discussion is August 9. It is very challenging to incorporate all these new writing and plotting into the main paper within a day without exceeding the 9-page limit required by NeurIPS.
> > >
> > > Again thank you for your review, and for helping us to improve the paper.

---

### Meta-Review · Area_Chair_jgFd · 2022-08-23

**Recommendation:** Accept
**Confidence:** Certain

**Metareview:**

Thank you for submitting your paper to NeurIPS! This paper makes a valuable contribution to the scoring model literature, providing a fast and scalable algorithm to derive sparse risk scores. The reviewers uniformly appreciated the methodological approach (integrating beam search with logistic regression, diverse feature selection, and star search for choosing integer coefficients), and noted that the stand-alone Python implementation is also advantageous over competitors that rely on mathematical programming solvers. I am pleased to recommend acceptance of this practically relevant work.

**Award:**

No

---

### Decision · Program_Chairs · 2022-09-14

Accept